# Decline of Sperm Quality over the Last Two Decades in the South of Europe: A Retrospective Study in Infertile Patients

**DOI:** 10.3390/biology12010070

**Published:** 2022-12-30

**Authors:** Emma Garcia-Grau, Judith Lleberia, Laura Costa, Miriam Guitart, Marc Yeste, Jordi Benet, María José Amengual, Jordi Ribas-Maynou

**Affiliations:** 1Departament of Obstetrics and Gynecology, Parc Taulí Health Corporation, ES-08208 Sabadell, Spain; 2Unit of Obstetrics and Gynecology, Department of Pediatrics, Obstetrics and Gynecology, and Preventive Medicine and Public Health, Faculty of Medicine, Autonomous University of Barcelona, Bellaterra, (Cerdanyola del Vallès), ES-08193 Barcelona, Spain; 3UDIAT Diagnostic Center, Parc Taulí Health Corporation, ES-08208 Sabadell, Spain; 4Biotechnology of Animal and Human Reproduction (TechnoSperm), Institute of Food and Agricultural Technology, University of Girona, ES-17003 Girona, Spain; 5Unit of Cell Biology, Department of Biology, Faculty of Sciences, University of Girona, ES-17003 Girona, Spain; 6Catalan Institution for Research and Advanced Studies (ICREA), ES-08010 Barcelona, Spain; 7Unit of Cell Biology and Medical Genetics, Department of Cell Biology, Physiology and Immunology, Faculty of Medicine, Autonomous University of Barcelona, Bellaterra (Cerdanyola del Vallès), ES-08193 Barcelona, Spain

**Keywords:** male infertility, morphology, progressive motility, semen quality, southern Europe, sperm

## Abstract

**Simple Summary:**

While previous studies showed that sperm quality has decreased over the last decades, others did not report such reductions, suggesting that the region where the population is evaluated has an influence. The current research, which was conducted on a regional scale, did not find variations in sperm count but did observe a drop in sperm motility and morphology.

**Abstract:**

Semen quality has a direct relation to male fertility. Whether sperm variables in humans have decreased over the last years is still uncertain, with some studies showing a decline and others reporting no changes. In this regard, previous research has suggested that lifestyle and environmental conditions may contribute to this variability, calling for regional studies. The present work is a retrospective, unicentric study that includes semen samples analyzed between 1997 and 2017 at the Parc Taulí Hospital (Barcelona metropolitan area). First, a multivariate analysis including the age as a confounding factor showed a statistically significant decrease in semen volume, pH, progressive motility, morphology and total motile sperm over time. Contrarily, no significant variation in sperm count or concentration was observed. Mean reductions per year were −0.02 mL for volume, −0.57% for progressively motile sperm and −0.72% for sperm with normal morphology. Interestingly, the average annual temperature registered by the Spanish Meteorology Agency negatively correlated to sperm morphology and sperm count (Rs = −0.642; *p* = 0.002 and Rs = −0.435; *p* = 0.049, respectively). In conclusion, the present study based on infertile patients from the Barcelona area found a decline in sperm motility and morphology, without effects on sperm count. Changes in temperature appeared to be associated to this decline, but further studies are needed to address the mechanisms linked to the observed variations.

## 1. Introduction

Infertility is a global public health issue that affects between 8% and 15% of couples of reproductive age [1]. It is well known that a male factor plays an important role in the disease, being present in at least half of infertile couples [2]. Thus, monitoring the evolution of semen parameters is crucial to define past, present and future trends of male reproductive capacity in humans, as accumulated reduction of sperm quality over the years may potentially end up in as severe issues in the near future.

One of the first meta-analyses on the topic published in the early 1990s showed a deterioration of sperm quality between 1938 and 1991 [3]. Years after this publication, the conclusions reached raised high interest and motivated other researchers to collect new data, some supporting [4,5,6,7,8,9,10,11,12,13,14] and others dismissing [15,16,17,18] a decline in sperm quality [15,16,17,18]. More recently, Levine et al. [14] conducted a meta-analysis based on 185 research articles from the previous 30 years, confirming a reduction of 50–60% in total sperm count between 1973 and 2011 (−2.23 million/year) in men from Western countries [14]; this work has recently been updated and obtained the same conclusions [19]. In spite of this, geographical differences were surmised to have some influence on the results, thus suggesting that further analyses at a regional scale should be conducted to reduce the inherent variability of the male population [14,19]. Aligned with this perspective, a recent systematic review of studies involving >300,000 healthy Chinese men showed a reduction in total sperm count of −2.07 million/year, between 1981 and 2019 [20]. On the other hand, other authors raised the bio-variability hypothesis, which contrasts with the previously reported sperm count decline, and states that variations in sperm count may result from non-pathological, species-specific causes [21]. According to Boulicault et al. [21], the causes alleged by Levine et al. [14] to explain the sperm count decline between 1973 and 2011 would need further confirmation, taking into account the potential flaws of the research literature compiled for the review [21].

While the debate on the evolution of semen quality during the last few decades is still open [22], the factors behind the drop in sperm count remain largely unknown. Related to this, it has been proposed that research should consider other variables, such as exposure to toxins or health status metrics, and conduct multiple testing in the same individuals over several years [21]. Different studies agree that sedentary lifestyles [23,24] and toxic habits, such as smoking [25,26], alcohol intake [27,28] and high fat diets [29], reduce sperm quality. Additionally, endocrine disruptors may negatively affect sperm production and quality through alterations in reproductive hormones [30,31], and air pollution has also been identified to affect spermatogenesis, leading to low sperm quality and DNA damage [32,33]. Finally, some authors proposed that environmental temperature also has repercussions on sperm quality [34] and advocated for investigations on a regional scale [14,19].

The aim of the present study was to provide further evidence on the decline of semen quality in infertile men at regional level. For this purpose, data from a specific population of infertile patients from the north-east of Spain were compiled over the last two decades, in order to bring additional clues on the aforementioned debate about the evolution of sperm parameters.

## 2. Materials and Methods

### 2.1. Study Design and Participants

The present study was designed as unicentric, retrospective study in the setting of the Parc Taulí Hospital, which belongs to the public health system and has an influence area of around 400,000 people in the Barcelona metropolitan area (Spain).

The data came from 8979 semen analyses performed between 1997 and 2017, conducted with the aid of a Computer-Assisted Sperm Analysis (CASA) system. The inclusion factors of the present study were: samples from infertile patients living in the north-east of Spain; individuals aged from 18 to 60 years old; and individuals attending the hospital for infertility reasons, either having normal or altered semen analysis. Infertility was defined as the condition of a couple that did not achieve a pregnancy after maintaining regular intercourse without contraception for 12 months. The exclusion factors were: repeated samples from the same patient; samples from patients taking any medication known to affect sperm quality; samples from patients attending the hospital for pathologies not related to infertility; and samples from patients aged >60 years old. As a result, data from the semen analyses of 5119 patients were finally included in the study.

### 2.2. Ethics Approval

Ethical approval for the present study was obtained from the Ethics Committee of Parc Taulí Hospital, Sabadell, Spain, prior to retrospective data collection (reference number: 2018565).

### 2.3. Semen Analysis

#### 2.3.1. Sperm Collection and Guidelines Followed

Semen samples were obtained by masturbation and ejaculation into a sterile plastic cup. Following physician recommendations, patients delivered a semen sample obtained by masturbation to the diagnostics laboratory within one hour post-ejaculation, the samples being collected in a sterile cup and preferably at hospital facilities. Immediately upon arrival, samples were kept at 37 °C for 30 min to allow liquefaction. At the time of sample delivery, sexual abstinence was confirmed to be between three and seven days, as a requirement for initiating semen analysis. The specific abstinence time, nevertheless, was not digitally recorded for all samples and therefore it could not be used in data analysis.

The analysis of the samples was performed in accordance to the World Health Organization guidelines [35,36], except for some points that are explained in the following sub-sections. Only raw percentages of the analyzed parameters were used. The diagnostic outcome based on WHO recommendations was not considered, given the changes in WHO reference values for sperm concentration, motility and morphology over the last two decades.

#### 2.3.2. Assessment of Macroscopic Parameters

In order to evaluate macroscopic parameters, semen samples were transferred into a test tube and thoroughly mixed. Although macroscopic evaluation included viscosity, color, volume and pH, only the two latter parameters were digitally recorded and could thus be included in the study. Volume was assessed by transferring the sample to a wide-bore volumetric tube and was recorded in milliliters. The pH was assessed by spreading a drop onto a pH paper (range 6.0 to 10.0) and comparing the uniform color obtained in the first 30 s with a calibration strip. These methods were conducted in accordance to the WHO laboratory manual for the examination and processing of human semen [35,36].

#### 2.3.3. Assessment of Sperm Concentration and Motility

Sperm concentration and motility were determined by pouring 10 µL of raw semen at 37 °C onto a pre-heated Makler chamber (Sefi-medical Instruments, Haifa, Israel). Immediately after, videos were recorded using a Computer-Assisted Sperm Analysis (CASA) system (Sperm Class Analyzer, SCA^®^, Microptic, Barcelona, Spain), whose software was updated from the first version (v1.0) to version 5 throughout the study. Videos were recorded at 10×, and included different fields. Samples were diluted in Ham F-10 Nutrient mixture medium when they were too concentrated (>200 sperm cells/per field). A visual inspection of fields to discard debris particles or other bright signals not corresponding to sperm cells was also conducted. If outcomes of the recorded fields differed in more than 10%, they were also discarded from the final analysis. A minimum of 400 sperm was required to evaluate concentration and motility; videos were captured until this figure was reached. The following motility parameters were recorded: percentage of sperm with progressive motility (type A + type B), percentage of sperm with non-progressive motility (type C) and percentage of immotile sperm (type D).

As an internal quality control, a pre-recorded video provided by the manufacturer was measured periodically. If deviations from the standards were observed, technical assistance was requested from the manufacturer, who restored software parameters. The Sperm Class Analyzer system was validated for the standardized analysis of human sperm in several published studies [37,38].

#### 2.3.4. Assessment of Sperm Morphology

For the evaluation of morphology, sperm were smeared on glass slides, air dried and stained with Papanicolaou dye. The samples were assessed using Kruger strict criteria [39] under a bright-field microscope at 1000× magnification by two different observers. When deviation between measures was higher than 10% of normal or abnormal forms, additional evaluations by an experienced researcher were made until the deviation was lower than 10%. When the deviation was lower than 10%, average values were calculated. The percentage of morphologically normal sperm was used for further analyses.

It is worth mentioning that the same experienced researcher trained all the observers involved in the microscopic assessment of sperm morphology. In addition, this researcher performed periodic checks on random slides to verify the accuracy of the morphology analysis; when deviations occurred, reanalysis was performed. Although these checks were not digitally recorded, fresh training to the specific observers was carried out if deviations occurred.

### 2.4. Recording the Average Temperature of Spain over the Period of Study

In order to get an approximation of the causes underlying the variations in sperm parameters, the average yearly temperature was retrieved from the National Meteorology Agency (AEMET), who provides publicly available climatological data. Specifically, data were obtained from a specific webpage of that Agency [40]. Average annual temperatures were tested for correlation to the yearly average sperm quality parameters (sperm concentration, total sperm count, progressive motility and morphology).

### 2.5. Statistical Analyses

Data analysis was performed using the SPSS software ver. 25.0 (Statistical Package for the Social Sciences, IBM Corp., Armonk, NY, USA). All variables included in this study were expressed as mean, standard deviation, median and interquartile range for the patients assessed each calendar year. The fitting of data to a normal distribution and homogeneity of variances was checked through Kolmogorov–Smirnov and Levene tests, respectively. Correlations between the mean of every parameter for each year were calculated through the Spearman test. A linear regression analysis was used to evaluate variations in the different semen parameters over the years. Finally, multiple linear regression models were run to identify which parameters varied over time, including the patient age as a confounding factor.

For all tests, the significance level was set at a *p*-value equal or less than 0.05 (95% of the confidence interval).

## 3. Results

### 3.1. Semen Quality Parameters

The average and standard deviation (SD) of semen parameters recorded between 1997 and 2017 were: 3.7 ± 1.8 mL for volume, 7.4 ± 0.3 for pH, 98 × 10^6^ ± 113 × 10^6^ sperm/mL for concentration, 340 × 10^6^ ± 392 × 10^6^ sperm for total sperm count, 157.9 × 10^6^ ± 236 × 10^6^ for motile sperm count, 38% ± 21% for sperm with progressive motility, 16% ± 10% for sperm with non-progressive motility, 45% ± 24% for immotile sperm, and 8% ± 8% for sperm with normal morphology. Age was 34 ± 6 years, and rates of azoospermia and cryptozoospermia were 4.55% and 0.64%, respectively.

A descriptive analysis of semen parameters (mean, standard deviation, median and interquartile range) by year is depicted in Table 1.

### 3.2. Variation of Each Parameter over Time

The changes in semen parameters were analyzed by multiple linear regression analysis. Table 2 shows the changes in semen parameters and the age of patients each year and those accumulated over the studied period with their corresponding *p*-values. Sperm concentration and total sperm count did not significantly vary throughout the period. In spite of this, statistically significant changes were observed for volume (−0.02 mL/year; *p* = 0.007), pH (−0.0064/year; *p* = 0.002), progressive motility (−0.57%/year; *p* = 0.022), non-progressive motility (+1.01%/year; *p* < 0.001), immotile sperm (−0.43%/year; *p* = 0.007) and normal morphology (−0.72%/year; *p* < 0.001). Interestingly, morphology showed a bimodal decreasing curve, with an annual reduction of −1.39% between 1997 and 2005, and an annual reduction of −0.24% between 2005 and 2017 (Figure 1).

Moreover, the age of infertile patients included in the study increased significantly (+0.08 years old/year).

### 3.3. Multivariate Analysis with the Age as a Confounding Factor

As age was previously demonstrated to be negatively correlated to routine semen parameters [41] and because mean male age was found to progressively increase in the patients included in this study (Table 2), a multivariate analysis including age as a confounding factor was conducted to identify whether the variation observed in the previous analysis could be biased by this parameter. This analysis revealed a significant decrease over time in the following variables: volume (β-coefficient: −0.145; 95% CI: −0.227 to −0.063, *p* < 0.001), pH (β-coefficient: −2.098; 95% CI: −2.597 to −1.599, *p* < 0.001), progressive motility (β-coefficient: −0.03; 95% CI: −0.037 to −0.023, *p* < 0.001), immotile sperm (β-coefficient: −0.029; 95% CI: −0.035 to −0.022, *p* < 0.001) and morphology (β-coefficient: −0.296; 95% CI: −0.313 to −0.280, *p* < 0.001). Conversely, non-progressive motility increased over time (β-coefficient: 0.271; 95% CI: 0.259 to 0.283, *p* < 0.001).

### 3.4. Relationship of Average Temperature with Sperm Count and Motility

Average temperatures were significantly correlated with the average percentage of sperm with normal morphology (Rs = −0.642; *p* = 0.001), and with the average total sperm count (Rs = −0.443; *p* = 0.048). In contrast, neither the percentage of sperm with progressive motility nor sperm concentration were correlated to the average temperature (*p* > 0.05).

## 4. Discussion

Sperm quality is essential to oocyte fertilization, and to give rise to healthy embryos and successful pregnancy [42,43]. In the context where human beings are exposed to several factors affecting their health, monitoring the sperm quality over time is crucial to prevent further impairment of men fertility. Additionally, as suggested before by different meta-analyses, this assessment should be conducted at a regional scale due to changes in lifestyle between communities [14,19]. The present retrospective, unicentric study aimed to determine the variation of sperm quality parameters in infertile men; for this purpose, single sperm samples from 5119 infertile patients attending a local hospital during a two-decade period were included in the study. These data were representative for the north-east of Spain and, specifically, the influence area of the Hospital was of around 400,000 people. As the age of the patients included statistically increased over the period of study, this variable was included as a confounding factor in a multivariate analysis model that ultimately demonstrated that semen volume, pH, progressive motility and normal morphology dropped during the last two decades.

At present, two opposing hypotheses about why sperm quality has declined in the general population coexist. The first hypothesis assumes that sperm quality is actively reduced due to lifestyle factors or exposure to environmental toxins and endocrine disruptors [14,20,44]. The second hypothesis posits that sperm quality naturally fluctuates over time [21]. Regardless of the premise, differences between the general population and infertile males may exist, and any attempt to elucidate whether environmental conditions affect sperm quality should distinguish between these two groups.

The present work found no significant decline in sperm concentration and total sperm count over the 20-year period. While these results are in agreement with a previous study that recruited 5000 (non-infertile) military draftees, they are in contrast with another study that observed a reduction of sperm count in southern Europe between 2001 and 2011 [8]. It is worth mentioning that whereas these two studies involved healthy, young individuals, the data of the present work were collected from infertile patients, which are known to present lower sperm count than healthy males [45]. One may, therefore, reasonably suggest that the differences in the effects of time on sperm quality between the two groups could come from the distinct condition of patients compared to healthy donors. This assumption, however, would still contrast with a larger study analyzing more than 25,000 infertile men, which showed a sperm concentration decrease at a rate of −1.9% per year [9]. A recent meta-analysis that included >300,000 Chinese men also noted a reduction of −2.07 million sperm per year [20]. It is known that geographical differences, which not only include exposure to toxins but also differences in lifestyle or diet, could also explain the inconsistencies between studies, including the current one [46,47,48]. Furthermore, separate investigations may not be absolutely comparable, as methodological differences and variations in the reference values set by the WHO inevitably affect how male infertility is diagnosed over time. This possibility has already been envisaged by other authors [49] and, since new recommendations are provided by the WHO [50], the impact of the proposed variations is now under scrutiny [51]. On the other hand, it is worth highlighting that sperm concentration in the current work was evaluated with the same computer-assisted sperm analysis system and the same Makler chamber model as was used during the research period. Moreover, the statistical analyses in the present work were conducted using the recorded values of sperm concentration and sperm count, rather than considering the WHO reference values utilized for diagnosis, which have varied over time [35,36]. All these methodological features could explain the differences between other studies and the present one.

Regarding sperm morphology, the results obtained in this work agree with previous reports, where a reduction in sperm morphology over time was also observed [8,10,12]. Specifically, an annual average decrease of −0.72% of sperm with normal morphology was observed. This reduction, notwithstanding, presented a bimodal distribution where it was −1.39% between 1997 and 2005, and decreased to only −0.24% between 2005 and 2017. Interestingly, this coincides with the higher reduction observed in the same period in a study conducted in infertile patients in France, which reported a decrease of −1.7% in the proportions of morphologically normal sperm [9]. Additionally, the research conducted in China by Huang et al. revealed an annual reduction of −1.4% in the proportions of morphologically normal sperm from semen donors between 2001 and 2015 [12].

A decline in sperm progressive motility of −0.57%/year was also observed in the current work. While this is in agreement with articles showing a reduction in this parameter during the recent decades, with annual reductions between −0.8% and −2.5% [12,52,53,54,55], other studies found no decrease in sperm progressive motility over time [8,56,57]. A decline in the percentage of immotile sperm (−0.43%/year) and an increase in non-progressive motile sperm (+1.01%/year) was also seen (Table 2). Whilst the increase in non-progressive motility could be explained by the reduction of progressive motility, other functional parameters, such as membrane integrity, calcium homeostasis and mitochondrial membrane potential, should be tested to provide a thorough explanation for the increase.

Finally, and in order to provide an insight into the causes leading to the sperm quality decline, the annual average temperature during the period of study was recorded and found to be negatively correlated with sperm morphology and sperm count. Spermatogenesis is a process that requires temperatures lower than the body [24,58]. Although thermoregulation allows the maintenance of testis temperature, heat stress is known to underlie a reduction in sperm quality [59]. In fact, different alterations arise from a substantial increase in testicular temperature, usually leading to lower sperm count and potentially affecting male fertility [60]. In spite of this, the impact of an increased environmental temperature on sperm quality has received less attention [34,61]. Based on the data compiled in the current work, it is reasonable to suggest that the environmental temperature could be one of the multifactorial causes leading to a reduction in sperm quality. However, the fact that, over the 20-year period, the temperature varied, regardless of other changes in lifestyle, should not be ruled out. For this reason, further research to confirm or dismiss this hypothesis is needed.

### Strengths and Limitations

The present work investigated the evolution of sperm quality taking into account age as a confounding factor and included only a single sample from each infertile patient. However, and in a similar fashion to other studies analyzing trends throughout a long period, the current one is not exempt of limitations. First, studies assessing External Quality Assessment (EQA) programs [62] demonstrated a high variability between different lab technicians in the evaluation of sperm morphology, which is associated with random or systemic errors. While a systematically reported quality control was not always applied during the current study, two measures by two different technicians were recorded; when >10% of variation was observed, the supervisor also analyzed the samples. Although multiple technicians analyzed samples, the unique lab supervisor who trained them reviewed the results in case of disparities.

Second, progressive motility resulted from the sum of ‘type a’ and ‘type b’ sperm, which has been shown to lead to much lower coefficients of variation [62]. Furthermore, lab technicians manually conducted the analyses and revised the videos captured with CASA after each assessment to remove the recordings that did not belong to sperm cells. Regardless of this, one cannot rule out inaccuracies. These possible inaccuracies and their impact on andrological assessment have been discussed in considerable detail by other authors [49] and are applicable to all studies assessing sperm quality.

Third, the present work used the Kruger strict criteria to determine sperm morphology [39]. Despite the changes proposed over the previous years to use a more accurate assessment [51,63,64,65], the evaluators followed the criteria published in 1993. Nonetheless, we cannot preclude that these evaluators were influenced by publications reporting these changes, although, as it has previously been stated in the Methods section, two lab technicians assessed samples in parallel and the same supervisor checked all the data.

Fourth, regarding methodological aspects, modifications in the reference values and recommendations given by the WHO could have an impact on the data, generating a bias between samples collected before and after the changes occurred (1999 and 2010). In this study, the same CASA software for the assessment of sperm concentration and motility was used (updated from v1 to v5), and the instrument was periodically calibrated. Although systems with automatic image analysis are not included in the WHO recommendations, automatic and manual analyses were previously compared without observing significant differences; in addition, the coefficients of variation for sperm concentration, motility and morphology are known to be lower in the case of automatic analysis [66].

In the current work, age was suggested to be a confounding factor for the changes in sperm quality over time. This study, nevertheless, lacks from specific lifestyle data coming from other potential confounding factors, such as smoking, alcohol drinking habits, body mass index or the exposure to harmful elements, that are known to affect semen quality. Dietary patterns, which also affect male fertility, should be included in future studies as a homogenizing factor to establish variations in different cohorts [67]. A special reference to the time of abstinence must be made, as it might modify certain sperm quality parameters [68]. In this study, abstinence was checked to be between three and seven days, but was not digitally recorded and was not thus available for statistical analysis.

Recent research showed that parameters such as sperm DNA fragmentation [69,70], miRNAs payload in sperm [71,72] and seminal plasma extracellular vesicles [73,74,75,76] could add complementary information to conventional semen analysis, thus obtaining a more comprehensive picture of sperm quality. Unfortunately, no historical data about these advanced sperm variables were available, so they could not be incorporated into this study.

Finally, while the current work suggests that temperature variations over time could be one of the factors implicated in the reduction of sperm quality, lifestyle conditions also changed during the period of study. For this reason, the possibility that the correlation observed resulted from a simultaneous variation of these factors should not be excluded. On the other hand, the present research included patients attending the hospital for infertility concerns; thus, the results compiled here cannot be extrapolated to the entire population.

## 5. Conclusions

This work supports a negative trend for sperm motility and morphology in the infertile population of the south of Europe, during the last two decades. In spite of this, the data collected from infertile men do not confirm the significant decline in sperm count reported in other studies. Further research should address the causes underlying the drop observed in sperm quality in the last two decades, as this may prevent further decreases.

## Figures and Tables

**Figure 1 biology-12-00070-f001:**
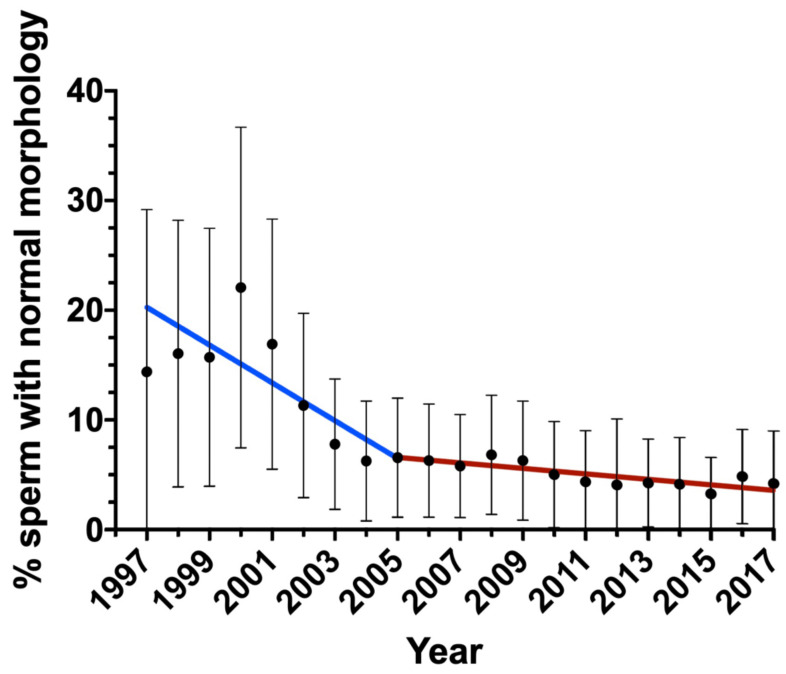
Change in the percentage of sperm with normal morphology during the period of study. Bars indicate standard deviation.

**Table 1 biology-12-00070-t001:** Semen parameters of the samples included in the study divided by year.

		Volume		pH		Sperm/mL		Total Sperm (× 10^6^)		Progressive Motility (%)		Non-Progressive Motility (%)		Immotile (%)		Normal Morphology (%)	
Year	n	Mean ± SD	Median (IQR)	Mean ± SD	Median (IQR)	Mean ± SD	Median (IQR)	Mean ± SD	Median (IQR)	Mean ± SD	Median (IQR)	Mean ± SD	Median (IQR)	Mean ± SD	Median (IQR)	Mean ± SD	Median (IQR)
1997	83	3.9 ± 1.6	3.5 (7.5)	7.5 ± 0.3	7.5 (1.6)	69 ± 60	58 (298)	286 ± 271	201 (1122)	50 ± 22	56 (90)	7 ± 5	6 (24)	44 ± 20	39 (81)	14 ± 15	8 (55)
1998	156	4.2 ± 2	4 (11.8)	7.4 ± 0.2	7.5 (1)	100 ± 95	83 (600)	405 ± 459	319 (3995)	49 ± 20	52 (85)	8 ± 6	6 (36)	44 ± 18	42 (87)	16 ± 12	13 (46)
1999	171	3.8 ± 1.7	3.5 (9.1)	7.4 ± 0.3	7.5 (1.6)	93 ± 95	55 (500)	315 ± 336	195 (2000)	43 ± 21	45 (88)	6 ± 6	4 (35)	51 ± 21	48 (85)	16 ± 12	14 (65)
2000	165	3.9 ± 1.8	3.8 (9.6)	7.4 ± 0.2	7.5 (1.3)	92 ± 87	75 (450)	343 ± 318	267 (1800)	50 ± 20	52 (90)	5 ± 6	4 (34)	44 ± 18	44 (89)	22 ± 15	20 (78)
2001	228	4 ± 1.8	3.8 (10)	7.4 ± 0.3	7.5 (2.3)	101 ± 121	70 (1000)	339 ± 316	262 (2160)	47 ± 22	47 (100)	4 ± 6	2 (33)	49 ± 20	50 (100)	17 ± 11	16 (57)
2002	258	3.8 ± 1.8	3.5 (8.5)	7.4 ± 0.3	7.5 (1.6)	86 ± 94	64 (600)	292 ± 298	219 (2000)	45 ± 24	48 (100)	4 ± 8	0 (44)	50 ± 21	49 (100)	11 ± 8	10 (44)
2003	210	3.7 ± 1.9	3.5 (9.3)	7.4 ± 0.3	7.5 (1.6)	103 ± 212	47 (2304)	326 ± 496	155 (4608)	40 ± 22	40 (88)	10 ± 10	8 (42)	48 ± 22	46 (100)	8 ± 6	7 (32)
2004	256	3.9 ± 1.7	3.8 (9.3)	7.4 ± 0.2	7.5 (1.3)	93 ± 87	73 (463)	348 ± 353	268 (2478)	38 ± 23	34 (85)	18 ± 9	17 (48)	45 ± 25	44 (98)	6 ± 5	5 (27)
2005	325	3.7 ± 1.8	3.5 (10.6)	7.4 ± 0.3	7.5 (1.6)	109 ± 147	70 (1134)	383 ± 550	234 (4575)	34 ± 20	32 (90)	17 ± 9	16 (68)	49 ± 24	48 (95)	7 ± 5	5 (42)
2006	297	3.7 ± 1.8	3.5 (10.4)	7.6 ± 0.4	7.5 (2)	102 ± 104	70 (657)	352 ± 365	246 (2481)	34 ± 19	32 (86)	19 ± 9	18 (50)	46 ± 23	46 (100)	6 ± 5	5 (38)
2007	272	4.1 ± 1.9	4 (15.5)	7.1 ± 0.3	7.2 (1.6)	123 ± ±120	95 (775)	480 ± 524	332 (3258)	35 ± 19	31 (86)	20 ± 8	20 (63)	45 ± 23	46 (100)	6 ± 5	5 (30)
2008	291	3.5 ± 1.8	3 (11.6)	7.3 ± 0.3	7.2 (1.6)	117 ± 120	84 (750)	393 ± 427	264 (3000)	36 ± 21	35 (100)	20 ± 9	20 (48)	43 ± 26	42 (100)	7 ± 5	6 (30)
2009	306	3.5 ± 1.7	3.2 (9.7)	7.4 ± 0.2	7.5 (1.6)	99 ± 101	70 (714)	309 ± 332	210 (2142)	31 ± 20	28 (86)	19 ± 9	18 (45)	50 ± 26	49 (100)	6 ± 5	5 (30)
2010	303	3.7 ± 1.8	3.5 (10)	7.2 ± 0.3	7.2 (1.3)	85 ± 90	59 (506)	275 ± 275	204 (1619)	30 ± 19	27 (83)	19 ± 9	19 (45)	50 ± 25	51 (91)	5 ± 5	4 (24)
2011	286	3.9 ± 2	3.6 (10.3)	7.3 ± 0.3	7.2 (1.6)	115 ± 171	70 (1624)	411 ± 599	245 (6496)	34 ± 20	31 (85)	21 ± 11	19 (69)	45 ± 25	45 (98)	4 ± 5	3 (30)
2012	258	3.7 ± 1.9	3.5 (9.6)	7.4 ± 0.3	7.2 (1.8)	94 ± 87	66 (537)	310 ± 297	205 (1500)	36 ± 22	33 (91)	23 ± 9	23 (60)	41 ± 25	42 (99)	4 ± 6	3 (74)
2013	304	3.7 ± 1.7	3.5 (12.4)	7.4 ± 0.4	7.2 (1.6)	84 ± 86	66 (875)	288 ± 306	197 (2584)	38 ± 21	35 (85)	21 ± 8	21 (44)	41 ± 25	40 (93)	4 ± 4	3 (26)
2014	280	3.6 ± 2	3.1 (15.6)	7.3 ± 0.3	7.2 (1.6)	80 ± 71	62 (352)	282 ± 318	195 (3168)	39 ± 21	39 (91)	21 ± 8	21 (47)	39 ± 24	35 (99)	4 ± 4	3 (23)
2015	220	3.8 ± 2	3.5 (11)	7.4 ± 0.3	7.2 (1.6)	77 ± 66	64 (423)	266 ± 243	180 (1203)	36 ± 20	36 (85)	22 ± 8	22 (40)	42 ± 23	40 (96)	3 ± 3	2 (16)
2016	241	3.5 ± 1.7	3.5 (7.6)	7.3 ± 0.3	7.2 (1.3)	115 ± 100	90 (541)	373 ± 357	284 (2528)	41 ± 22	42 (90)	23 ± 8	23 (46)	36 ± 24	31 (95)	5 ± 4	4 (24)
2017	208	3.7 ± 1.8	3.5 (9.8)	7.4 ± 0.3	7.5 (1.3)	93 ± 90	69 (569)	325 ± 341	209 (2325)	40 ± 22	40 (91)	23 ± 8	24 (38)	37 ± 24	35 (99)	4 ± 5	3 (40)

IQR: Interquartile range; SD: Standard deviation.

**Table 2 biology-12-00070-t002:** Coefficients of correlation (Rs) between time and yearly mean of each semen variable, and changes of means in one year and after 21 years.

Parameter	Rs	*p*-Value	Mean Change in One Year	Mean Change after 21 Years
Volume	−0.571	0.007	−0.02 mL	−0.42 mL
pH	−0.630	0.002	−0.0064	−0.1344
Sperm concentration	−0.009	0.969		
Total sperm count	−0.195	0.397		
Progressive motility (%)	−0.496	0.022	−0.57%	−11.97%
Non-progressive motility (%)	0.925	<0.001	+1.01%	+21.21%
Immotile sperm (%)	−0.570	0.007	−0.43%	−9.03%
Total progressive motile sperm	−0.392	0.079	−2.33 × 10^6^ sperm	48.93 × 10^6^ sperm
Morphology (% normal)	−0.910	<0.001	−0.72%	−15.12%
Age	0.649	<0.001	+0.08 years	+1.68 years

## Data Availability

Data are available upon request to the authors.

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
