# Peer review of "Decline of Sperm Quality over the Last Two Decades in the South of Europe: A Retrospective Study in Infertile Patients"

_biology, 2022, doi:10.3390/biology12010070_

Round 1

Reviewer 1 Report (Previous Reviewer 3)

Thank you for allowing me to re-review the manuscript: "Decline of sperm quality over the last two decades in the south of Europe: a retrospective study in infertile patients". The aims of the authors was to study temporal trends in sperm parameters over 20years in infertile patients at a single center I think the authors did a good job in answering my previous comments Please note - that the meta-regression analysis by levine that you cited has been updates and perhaps this update should be cited rather than the previous article (the conclusion remains the same, thus is should not affect the rest of your manuscript) Levine H, Jørgensen N, Martino-Andrade A, et al. Temporal trends in sperm count: a systematic review and meta-regression analysis of samples collected globally in the 20th and 21st centuries [published online ahead of print, 2022 Nov 15]. /Hum Reprod Update/. 2022;dmac035. doi:10.1093/humupd/dmac035 While some grammatical editing may be needed I have no other comments.

Author Response

Dear Editors and Reviewers,

Enclosed please find the revised manuscript No. biology-2109743 entitled “Decline of sperm quality over the last two decades in the south of Europe: a retrospective study in infertile patients”, which we would like to be reconsidered for publication in Biology.

We sincerely thank the Editors and Reviewers for the opportunity to get our manuscript reviewed for the second time, and for their valuable and constructive comments. We have tried our best to address in detail all the concerns and questions raised by the Reviewers. Here, we provide itemized responses to each point raised by each of the Reviewers and we made the changes (track changes mode in MS Word) to the Manuscript, when required.

We hope that the improved version of our work will now be suitable for publication in Biology.

Dr. Jordi Ribas Maynou on behalf of all Authors.

Reviewer 1

Comment: “Thank you for allowing me to re-review the manuscript: "Decline of sperm quality over the last two decades in the south of Europe: a retrospective study in infertile patients". The aims of the authors was to study temporal trends in sperm parameters over 20years in infertile patients at a single center I think the authors did a good job in answering my previous comments

Please note - that the meta-regression analysis by levine that you cited has been updates and perhaps this update should be cited rather than the previous article (the conclusion remains the same, thus is should not affect the rest of your manuscript):

Levine H, Jørgensen N, Martino-Andrade A, et al. Temporal trends in sperm count: a systematic review and meta-regression analysis of samples collected globally in the 20th and 21st centuries [published online ahead of print, 2022 Nov 15]. /Hum Reprod Update/. 2022;dmac035. doi:10.1093/humupd/dmac035

While some grammatical editing may be needed I have no other comments.”

Authors’ answer: Thank you very much for re-reviewing our manuscript. We have now included the new work by Levine et al., and we have conducted an extensive proofreading and revision of the English/written quality of the Manuscript.

Reviewer 2 Report (New Reviewer)

The study is interesting but many questions must be answered by the authors...title is good 

Abstract delete manu lines in thr beginning and summarize those lines in 2 lines only... conclusion of the abstract is  very bad in writing please reorganize 

Keywords should be placed in alphabetical order...

What about references present in the semen collection and macroscopic analysis ..this part must contain anew references....what about inclusion and exclusion criteria for this study.....iam not understanding the stastical test please reorganize this paragraph to be more clear....in the result section ..this part should be summarized as possible

..what about absences of errors bars in figure 1 please justify... what about future aspects ???.

Author Response

Dear Editors and Reviewers,

Enclosed please find the revised manuscript No. biology-2109743 entitled “Decline of sperm quality over the last two decades in the south of Europe: a retrospective study in infertile patients”, which we would like to be reconsidered for publication in Biology.

We sincerely thank the Editors and Reviewers for the opportunity to get our manuscript reviewed for the second time, and for their valuable and constructive comments. We have tried our best to address in detail all the concerns and questions raised by the Reviewers. Here, we provide itemized responses to each point raised by each of the Reviewers and we made the changes (track changes mode in MS Word) to the Manuscript, when required.

We hope that the improved version of our work will now be suitable for publication in Biology.

Dr. Jordi Ribas Maynou on behalf of all Authors.

Reviewer: 2

Comment 1: “The study is interesting but many questions must be answered by the authors...title is good”

Authors’ answer: First of all, authors would like to thank the Reviewer for the revision made on our Manuscript and for the positive comments, which definitely helped us to improve our work. Below, we address the suggestions raised by the Reviewer on a point-by-point basis, and we included the changes using the MS-word Track Changes function.

Comment 2: “Abstract delete manu lines in thr beginning and summarize those lines in 2 lines only... conclusion of the abstract is  very bad in writing please reorganize”

Authors’ answer: We thank the Reviewer for the comment. As per their request, we have rewritten the first part of the Abstract, shortening its length and providing a more direct, clear message. Also, we have revised and rephrased the conclusions of the Abstract.

Comment 3: “Keywords should be placed in alphabetical order...”

Authors’ answer: We thank the Reviewer for drawing our attention into this. We have reordered the keywords, as per the Reviewer’s request.

Comment 4: “What about references present in the semen collection and macroscopic analysis ..this part must contain anew references....”

Authors’ answer: Thanks for the comment. Semen collection was described in Section 2.3.1, and macroscopic parameters were detailed in section 2.3.2. That being said, we are willing to accept, after reading the Reviewer’s comment, that, perhaps, not enough details were provided. Thus, we have revised the sections to describe in more detail the methods. In addition, we have made reference to the World Health Organization laboratory manual for the examination and processing of human semen, which stated the protocol to conduct their macroscopic evaluation.

Comment 5: “what about inclusion and exclusion criteria for this study.....”

Authors’ answer: We thank the Reviewer for raising this important question. We have revised the Manuscript in order to ensure that both inclusion and exclusion criteria were clearly described in detail in Section 2.1. These criteria were:

  • Inclusion:
  • Infertile patients living in north-east of Spain
  • Age between 18 and 60 years old
  • Attending the hospital for infertility reasons, either having normal or altered semen analysis.
  • Exclusion:
  • Repeated samples from the same patient
  • Samples from patients taking medications that affect sperm quality
  • Patients attending the hospital for different reasons than infertility.

Comment 6: “iam not understanding the stastical test please reorganize this paragraph to be more clear....”

Authors’ answer: Thanks for pointing that statistical methods needed clarification. Following the reviewer’s request, we have tried our best to add as much detail as we could to the statistical methods (section 2.5). As we have stated in this section, we calculated the mean of each parameter (sperm count, motility and morphology) for each year, and we worked out the correlation between sperm parameters and time. Then, in order to establish the variation for each year, we conducted a linear regression analysis. Finally, since patient age varied along the years, we ran multiple regression models and the patient age was included as a confounding factor.

Comment 7: “in the result section ..this part should be summarized as possible”

Authors’ answer: Thanks for the suggestion. We have tried to be as succinct as possible, and results have been presented in two tables, which include most of the collected information and data. Also, in Section 3.3 we have summarized which variables still maintained the variation after including the male age as a confounding factor. In addition to this, we have gone through the Manuscript again to try summarize all the results as much as possible; we do honestly believe, as it stands in this current revised version, we cannot summarize further. Finally, we have made some editorial/grammar corrections on this section to improve its readability. We hope that the Reviewer, after making all these improvements, will consider this section appropriate.

Comment 8: “..what about absences of errors bars in figure 1 please justify... what about future aspects ???”

Authors’ answer: We thank the Reviewer for this comment. This confusion could be caused by the fact that Standard Error of the Mean (SEM) rather than Standard Deviation (SD) were represented, the former being smaller than the latter, which could explain why the Reviewer found that bars were very small. Following the Reviewer’s comment and in order to fully address the concern, a new Figure 1 including Standard Deviation has been prepared. This may, indeed, be more representative of the nature/distribution of our data.

Reviewer 3 Report (New Reviewer)

This article supports a negative trend for sperm motility and morphology, ONLY in the infertile population from the region of Spain. (at the Parc TaulíHospital (Barcelona metropolitan area).

Some information about the inclusion and exclusion criteria of articles and subjects should also be provided.

The literature also mentions the age limits of the patients, the human race, the phenomenon of migration... Information should be provided about the lifestyle of the subjects, risks, vices, etc.

To the material and method.

It is hard to believe that all 8,979 sperm samples performed between 1997 and 2017 were evaluated with the CASA system. Were they even made in 1997?

Like limitations:

-The article is based on some common, general sperm determinations that have been evaluated by slightly used, although not wrong, methods. The manuscript seems to have technical and andrological nomenclature mistakes (see the yellow notes in the attachment).

-Why wasn't an automatic Ph-meter used?

-Seminal plasma contains numerous extracellular vesicles (sEVs) apparently involved in male (in)fertility. Another limitation of the study would be the evaluation of Extrcelular Vesicles in the seminal fluid. It is still a research article and the latest recommendations in the field must be taken into account. I am attaching some articles in this regard:

Abu-Halima, M.; Ludwig, N.; Hart, M.; Leidinger, P.; Backes, C.; Keller, A.; Hammadeh, M.; Meese, E. Altered Micro-Ribonucleic Acid Expression Profiles of Extracellular Microvesicles in the Seminal Plasma of Patients with Oligoasthenozoospermia. Fertil. Steril. 2016, 106, 1061-1069.e3

Kumar, N.; Singh, N.K. “Emerging Role of Novel Seminal Plasma Bio-Markers in Male Infertility: A Review.” Eur. J. Obstet. Gynecol. Reprod. Biol. 2020, 253, 170–179

Hong, Y.; Wu, Y.; Zhang, J.; Yu, C.; Shen, L.; Chen, H.; Chen, L.; Zhou, X.; Gao, F. Decreased PiRNAs in Infertile Semen Are Related to Downregulation of Sperm MitoPLD Expression. Front. Endocrinol. (Lausanne). 2021, 12, 696121

Since 2014 EVs characterization is mandatory as reported in the MISEV guidelines (Lötvall J et al. Minimal experimental requirements for definition of extracellular vesicles and their functions: a position statement from the International Society for Extracellular Vesicles. 2014;3:26913. doi: 10.3402/jev.v3.26913. eCollection 2014. Thery C, et al. Minimal information for studies of extracellular vesicles 2018 (MISEV2018): a position statement of the international society for extracellular vesicles and update of the MISEV2014 guidelines. J Extracell Vesicles. 2018;7:1535750) and required a combination of electronic microscopic evaluation and western blot analysis with specific EVs marker antibodies.

Author Response

Dear Editors and Reviewers,

Enclosed please find the revised manuscript No. biology-2109743 entitled “Decline of sperm quality over the last two decades in the south of Europe: a retrospective study in infertile patients”, which we would like to be reconsidered for publication in Biology.

We sincerely thank the Editors and Reviewers for the opportunity to get our manuscript reviewed for the second time, and for their valuable and constructive comments. We have tried our best to address in detail all the concerns and questions raised by the Reviewers. Here, we provide itemized responses to each point raised by each of the Reviewers and we made the changes (track changes mode in MS Word) to the Manuscript, when required.

We hope that the improved version of our work will now be suitable for publication in Biology.

Dr. Jordi Ribas Maynou on behalf of all Authors.

Reviewer 3

Comment 1: “This article supports a negative trend for sperm motility and morphology, ONLY in the infertile population from the region of Spain. (at the Parc TaulíHospital (Barcelona metropolitan area).”

Authors’ answer: First of all, authors would like to thank the Reviewer for the revision made on our Manuscript and for their comments, which have definitely improved the quality of the work. Below, we address the suggestions raised on a point-by-point basis. Changes to the Manuscript have been made using the MS-word Track Changes function. 

The Reviewer is right that our study is limited to the region of influence of Parc Taulí Hospital, and that it only included the infertile population. Although, at first glance, this could seem a limitation, it is important to bear in mind that previous meta-analyses conducted in the topic encouraged further studies to be conducted at regional scale, in order to reduce the variability inherent to the different lifestyle conditions and environmental variations.

Comment 2: “Some information about the inclusion and exclusion criteria of articles and subjects should also be provided.”

Authors’ answer: We thank the Reviewer for this suggestion. While we certainly indicated, in the previous version, that the study was performed on a regional scale and included infertile patients, we have provided a more detailed explained on inclusion and exclusion criteria in the revised version (Section 2.1). These criteria were:

  • Inclusion:
  • Infertile patients living in north-east of Spain
  • Age between 18 and 60 years old
  • Attending the hospital for infertility reasons, either having normal or altered semen analysis.
  • Exclusion:
  • Repeated samples from the same patient
  • Samples from patients taking medications that affect sperm quality
  • Patients attending the hospital for different reasons than infertility.
  • Samples from patients more than 60 years old.

Comment 3: “The literature also mentions the age limits of the patients, the human race, the phenomenon of migration... Information should be provided about the lifestyle of the subjects, risks, vices, etc.”

Authors’ answer: We thank the Reviewer for the comment. Regarding inclusion criteria, patients had to be between 18 and 60 years old (section 2.1). As exclusion criteria, we excluded those patients that were taking any medication known to affect semen quality. The Reviewer is correct that other data about lifestyle conditions would be important to establish more homogeneous cohorts. Unfortunately, our study is of retrospective nature, and these data were not recorded when semen samples were collected and analyzed in the laboratory. As we agree with the Reviewer that it would have been good if these data were available, the Discussion has been revised in order to emphasize this need/limitation (lines 636-638)

Comment 4: “To the material and method.

Line 276: It is hard to believe that all 8,979 sperm samples performed between 1997 and 2017 were evaluated with the CASA system. Were they even made in 1997?”

Authors’ answer: Thanks for the comment. We are sure if we get what the reviewer intended to mean. If we were not wrong, the reviewer was skeptical on whether the Hospital evaluated sperm motility with a CASA system. Yet, and said with all due respect, we do not understand why the Reviewer does not believe that a public hospital with a dedicated Department of Obstetrics and Gynecology and an integrated Diagnostic Center routinely assesses sperm quality, because these data are of high importance for clinicians to define possible male infertility causes and define further infertility treatments. The Reviewer could take into account that the present is a retrospective study including historical and recorded data of sperm quality assessed with the CASA system (the same system with the corresponding software updates (from v.1 to v.5, as specified in section 2.3.3). CASA systems were already used in 1997, and there are several research works from that time applying this type of sperm analysis in humans and animals. To mention some, please find the following references of studies conducted between 1997 and 1998:

10.1093/humrep/13.9.2512

10.1093/molehr/4.5.439.

10.1095/biolreprod58.6.1515

10.1016/s0093-691x(98)00110-1.

10.1016/S0093-691X(98)00069-7

10.1046/j.1365-2605.1997.00072.x.

10.1046/j.1365-2605.1998.00138.x

Comment 5: “Like limitations:

Line 218: The article is based on some common, general sperm determinations that have been evaluated by slightly used, although not wrong, methods. The manuscript seems to have technical and andrological nomenclature mistakes (see the yellow notes in the attachment).”

Authors’ answer: Thanks for noting these points in the attachment, we answer them point-by-point below.

Comment 6: “Line 269: Why wasn't an automatic Ph-meter used?

Authors’ answer: Thanks for asking this. A human ejaculate has a 3.7 ± 1.8 mL of volume, which limits using a pH-meter due to the low volume. Importantly, using a pH-meter would be a source of cross-contamination in a diagnostics laboratory conducting basic semen analysis. These are the reasons of utilizing pH paper/strips instead. Also, one should also note that using pH-paper is established as the standard methodology in WHO Laboratory manual for the Examination and Processing of Human Semen. Thanks to the comment of the Reviewer, we included a sentence referencing to the standard method in Section 2.3.2.

Comment 7: “Line 281: I think that you can use the term - extended semen more correctly.”

Authors’ answer: Thanks for pointing this out. The term “extended” semen is a term used in veterinary medicine, rather than in human medicine. In human reproduction laboratories, samples are not directly diluted in diluents, but treated with sperm washing medium and processed through swim-up or density gradients. In the sentence the Reviewer is referring to, we were stating that samples were diluted in culture media in order to reduce sperm concentration, providing a more accurate measurement of sperm motility and concentration in the CASA system. This is the usual practice in human andrology.

Comment 8: “Line 483: “to oocyte fertilization, and produce healthy embryos”

Authors’ answer: We thank the Reviewer for their comment. The sentence proposed has been included.

Comment 9: “-Seminal plasma contains numerous extracellular vesicles (sEVs) apparently involved in male (in)fertility. Another limitation of the study would be the evaluation of Extrcelular Vesicles in the seminal fluid. It is still a research article and the latest recommendations in the field must be taken into account. I am attaching some articles in this regard:

Abu-Halima, M.; Ludwig, N.; Hart, M.; Leidinger, P.; Backes, C.; Keller, A.; Hammadeh, M.; Meese, E. Altered Micro-Ribonucleic Acid Expression Profiles of Extracellular Microvesicles in the Seminal Plasma of Patients with Oligoasthenozoospermia. Fertil. Steril. 2016, 106, 1061-1069.e3

Kumar, N.; Singh, N.K. “Emerging Role of Novel Seminal Plasma Bio-Markers in Male Infertility: A Review.” Eur. J. Obstet. Gynecol. Reprod. Biol. 2020, 253, 170–179 

Hong, Y.; Wu, Y.; Zhang, J.; Yu, C.; Shen, L.; Chen, H.; Chen, L.; Zhou, X.; Gao, F. Decreased PiRNAs in Infertile Semen Are Related to Downregulation of Sperm MitoPLD Expression. Front. Endocrinol. (Lausanne). 2021, 12, 696121 

Since 2014 EVs characterization is mandatory as reported in the MISEV guidelines (Lötvall J et al. Minimal experimental requirements for definition of extracellular vesicles and their functions: a position statement from the International Society for Extracellular Vesicles. 2014;3:26913. doi: 10.3402/jev.v3.26913. eCollection 2014. Thery C, et al. Minimal information for studies of extracellular vesicles 2018 (MISEV2018): a position statement of the international society for extracellular vesicles and update of the MISEV2014 guidelines. J Extracell Vesicles. 2018;7:1535750) and required a combination of electronic microscopic evaluation and western blot analysis with specific EVs marker antibodies”

Authors’ answer: We thank the Reviewer for mentioning this issue and making this suggestion. We agree with the Reviewer that advanced methodologies could add useful information to determine the trends sperm quality is following. While some methods were not, unfortunately, available 20 years ago and therefore no historical data could be recorded, we have revised the Manuscript and included the suggestion of the Reviewer as well as the references in the limitations section.

Round 2

Reviewer 2 Report (New Reviewer)

The paper is  now improved.... lines 114-122need to  e rephrased again ....thanks 

Author Response

The paper is  now improved.... lines 114-122need to  e rephrased again ....thanks 

Response:  This paragraph has been revised as per the reviewer's request.

This manuscript is a resubmission of an earlier submission. The following is a list of the peer review reports and author responses from that submission.

Round 1

Reviewer 1 Report

This manuscript is quite well organized, the authors have argued the purpose of the research against the background of the current literature. The discussion clearly discussed the results, confronting them with the latest reports on the subject. However, there are some gaps that need to be filled.

Main Comments:

L115-129: Little is known about the patients from whom the samples were taken. Describe:

- whether and how they were grouped,

- what age range they were in,

- where did they come from (origin)

- whether there was a history of disease burden,

- hygiene of life - smoking, alcohol consumption,

- what was their diet like, were they taking any treatment?

You should definitely include this information in the M&M section. For clarity, the authors may place the data in a table. Their presentation is necessary for a better understanding of the research, especially since the authors referred to the age of the examined men in the further part of this manuscript.

Moreover, the samples were taken over a 20-year period, during which the availability and precision of semen analysis methods increased significantly, so the method of sample collection and preparation may have changed. It is also important whether the analyzes were performed by the same team throughout the period or by different people. These issues should be taken into account.

L142: specify the software version (the oldest possible version was used for 20 years?)

Fig 1 should be moved to the results section

Reviewer 2 Report

The authos  Garcia-Grau et al. presented a paper titled “Decline of sperm quality over the last two decades in the south of Europe: a retrospective study in infertile patients.

The topic is interesting however important issues have to be clarified.

The patient’s selection is not well defined, and this point is of primary importance. It is not clear the infertility cause that determine this condition. The different causes associated with male infertility have to be considered.

The discussion is not focalized on the results, confused. Some inaccuracies are present regarding the varicocele condition and environmental influence.

English language should be revised.

Minor remarks

The organization of references in the text and at the end have to be revised.

Key words- change morphology with sperm morphology.

2.3 Semen analysis….the numbering of subsequent parameters must be correct

Reviewer 3 Report

Thank you for allowing me to review the manuscript: "Decline of sperm quality over the last two decades in the south of Europe: a retrospective study in infertile patients".

The aims of the authors was to study temporal trends in sperm parameters over 20 years in infertile patients at a single center.

While this topic has been studies a lot, it is still of interest and of overall merit. For example just last year this has been the topic of a fertility debate: Jørgensen N, Lamb DJ, Levine H, et al. Are worldwide sperm counts declining?. Fertil Steril. 2021;116(6):1457-1463. doi:10.1016/j.fertnstert.2021.10.020

The strengths of this article stem from it being a single center study with documented methods for semen analysis making analysis and comparison of the results more relevant.

Two minor recommendations:

- Although the authors state using the Kruger strict criteria for assessing sperm morphology a possible limitation that must be addressed is that these criteria too have been modified over the years:

Wald G, Punjani N, Hayden R, Feliciano M, Dudley V, Goldstein M. Assessing the clinical value of the Kruger strict morphology criteria over the World Health Organization fourth edition criteria. F S Rep. 2021;2(2):176-180. Published 2021 Apr 19. doi:10.1016/j.xfre.2021.04.003

Menkveld R, Rhemrev JP, Franken DR, Vermeiden JP, Kruger TF. Acrosomal morphology as a novel criterion for male fertility diagnosis: relation with acrosin activity, morphology (strict criteria), and fertilization in vitro. Fertil Steril. 1996;65(3):637-644. doi:10.1016/s0015-0282(16)58167-9

There is also a known tendency for stricter approach over time when assessing sperm morphology: 

Menkveld R. Clinical significance of the low normal sperm morphology value as proposed in the fifth edition of the WHO Laboratory Manual for the Examination and Processing of Human Semen. Asian J Androl. 2010;12(1):47-58. doi:10.1038/aja.2009.14

These changes (changes in the criteria and a stricter approach in assessing sperm parameters) may have, at least partially, contributed to the decline in sperm morphology noted over the study period.

Indeed, in another single center study a decline was noted when stricter criteria for assessing sperm were adopted. In that same study no significant changes in other sperm parameters where noted. 

Feferkorn I, Shrem G, Azani L, et al. Hope for male fecundity: clinically insignificant changes in semen parameters over 10 years at a single clinic while assessing an infertility population [published correction appears in J Assist Reprod Genet. 2021 Sep 13;:]. J Assist Reprod Genet. 2021;38(11):2995-3002. doi:10.1007/s10815-021-02298-8

Perhaps the factor that is least susceptible to change in method of assessment is sperm concentration. Indeed, no change was found in this parameter over the study period.

Second:

The suggestion that environmental temperature is associated with the decline in some of the sperm parameters, while, as stated by the authors, has some biological plausibility, in the case of this study the association can be coincidental as many environmental factors (and other factors as well) have  changed over the time period. While the authors acknowledge that further study is needed, the limitation in the basis of this assumption (of a possible cause effect relationship) must be, in my opinion, emphasized. 

It is also worth noting that most studies on the association between temperature and sperm parameters describe a decline in sperm concentration - a decline not found in this study, again putting this relationship between environmental temperature and the decline in sperm parameters into question.

Apart from these minor suggestion I find this study of merit and worthy of publications.